# Delta-Radiomics Biomarker in Colorectal Cancer Liver Metastases Treated with Cetuximab Plus Avelumab (CAVE Trial)

**DOI:** 10.3390/diagnostics15222914

**Published:** 2025-11-18

**Authors:** Valerio Nardone, Vittorio Patanè, Luca Marinelli, Luca D’Ambrosio, Sara Del Tufo, Marco De Chiara, Maria Chiara Brunese, Dino Rubini, Roberta Grassi, Anna Russo, Maria Paola Belfiore, Fortunato Ciardiello, Salvatore Cappabianca, Erika Martinelli, Alfonso Reginelli

**Affiliations:** Precision Medicine Department, University of Campania “Luigi Vanvitelli”, Piazza Luigi Miraglia 2, 80138 Naples, Italy; valerio.nardone@unicampania.it (V.N.); mariner189@gmail.com (L.M.); luca.dambrosio@studenti.unicampania.it (L.D.); sara.deltufo95@gmail.com (S.D.T.); mariachiara.brunese@unicampania.it (M.C.B.); dino.rubini@studenti.unicampania.it (D.R.); roberta.grassi@policliniconapoli.it (R.G.); annarusso81@yahoo.it (A.R.); mariapaola.belfiore@unicampania.it (M.P.B.); fortunato.ciardiello@unicampania.it (F.C.); salvatore.cappabianca@unicampania.it (S.C.); erika.martinelli@unicampania.it (E.M.); alfonso.reginelli@unicampania.it (A.R.)

**Keywords:** delta-radiomics, CT texture analysis, metastatic colorectal cancer, liver metastases, immunotherapy, EGFR rechallenge, precision oncology, radiomics

## Abstract

**Background**: Radiomics enables the extraction of quantitative imaging biomarkers that can non-invasively capture tumor biology and treatment response. Delta-radiomics, by assessing temporal changes in radiomic features, may improve reproducibility and reveal early therapy-induced alterations. This study investigated whether delta-texture features from contrast-enhanced CT could predict progression-free survival (PFS) and overall survival (OS) in patients with metastatic colorectal cancer (mCRC) liver metastases treated with cetuximab rechallenge plus avelumab within the CAVE trial. **Methods**: This retrospective substudy included 42 patients enrolled in the multicenter CAVE phase II trial with evaluable liver metastases on baseline and first restaging CT. Liver lesions were manually segmented by two readers, and radiomic features were extracted according to IBSI guidelines. Delta-values were calculated as relative changes between baseline and post-treatment scans. Reproducibility (ICC > 0.70), univariate and multivariable analyses, ROC/AUC, bootstrap resampling, cross-validation, and decision curve analysis were performed to evaluate predictive performance and clinical utility. **Results**: Among reproducible features, delta-GLCM Homogeneity emerged as the most robust predictor. A decrease in homogeneity independently correlated with longer PFS (HR = 0.32, *p* = 0.003) and OS (HR = 0.41, *p* = 0.021). The combined clinical–radiomic model achieved good discrimination (AUC 0.94 training, 0.74 validation) and stable performance on internal validation (bootstrap C-index 0.77). Decision curve analysis indicated greater net clinical benefit compared with clinical variables alone. **Conclusions**: This exploratory study provides preliminary evidence that delta-GLCM Homogeneity may serve as a reproducible imaging biomarker of response and survival in mCRC patients receiving cetuximab plus avelumab rechallenge. If validated in larger, independent cohorts, delta-radiomics could enable early identification of non-responders and support personalized treatment adaptation in immuno-targeted therapy. Given the small sample size, the potential for overfitting should be considered. Future work should prioritize prospective multicenter validation with a pre-registered, locked model and explore multimodal integration (radiogenomics, circulating biomarkers, and AI-driven fusion of imaging with clinical/omic data) to strengthen translational impact. Beyond imaging advances, these findings align with broader trends in personalized oncology, including response-adaptive strategies, multimodal biomarker integration, and AI-enabled decision support.

## 1. Introduction

Radiomics, the high-throughput extraction of quantitative descriptors from standard medical images, is increasingly recognized as a valuable tool in precision oncology [1,2,3]. By transforming routinely acquired scans into quantitative data, radiomics can uncover imaging biomarkers that complement histopathologic and molecular information [4,5,6,7,8]. While radiologists interpret visual patterns, many biologically relevant details remain imperceptible to the human eye. Radiomic analysis quantifies tumor morphology, intensity, and texture, offering a reproducible way to characterize intratumoral heterogeneity—a hallmark of cancer associated with treatment response and survival [9,10,11,12,13].

Texture analysis (TA), a core component of radiomics, measures spatial variations in gray-level intensity that reflect biological complexity, including angiogenesis, stromal remodeling, necrosis, and cellular proliferation [4,12,14,15,16]. Increased heterogeneity has been associated with tumor aggressiveness, resistance to therapy, and distinct molecular profiles across several malignancies [17,18,19,20]. These links support the concept of imaging phenotypes that mirror the underlying tumor biology.

Within this framework, delta-radiomics—which evaluates temporal changes in radiomic features between sequential imaging studies—has emerged as a particularly robust approach [21,22,23,24]. By focusing on intra-patient variation rather than absolute values, delta-radiomics mitigates the influence of acquisition parameters and scanner differences, enhancing reproducibility in multicenter settings. Moreover, it can detect subtle, therapy-induced structural changes—such as necrosis or stromal alterations—that often precede measurable tumor shrinkage. These early variations could provide a window for adapting therapy before conventional criteria, such as RECIST, indicate progression or response [22,25,26,27].

Colorectal cancer (CRC) is the third most commonly diagnosed malignancy worldwide and a leading cause of cancer-related death [1,28,29]. The liver represents the predominant site of metastasis, and outcomes for patients with metastatic colorectal cancer (mCRC) remain highly variable despite therapeutic advances [30,31,32,33]. For RAS wild-type disease, anti-EGFR monoclonal antibodies provide clinical benefit, but resistance invariably develops [34,35]. Rechallenge strategies with EGFR inhibitors, especially in combination with immunotherapy, are being explored to prolong disease control [36,37,38,39,40]. However, there are currently no validated imaging biomarkers capable of identifying patients who may benefit from these regimens at an early stage.

Response evaluation in mCRC largely relies on RECIST 1.1 criteria, based on unidimensional measurements [41,42,43]. Although widely used, RECIST may underestimate the biological effects of targeted or immune-based treatments, which often induce changes in tumor composition rather than size [44,45,46]. This limitation highlights a pressing unmet need for non-invasive, reproducible, and clinically actionable biomarkers that can capture early biological responses.

Radiomics has gained clinical traction across the oncology continuum—supporting tumor staging, early diagnosis and differential diagnosis, prognosis prediction, and treatment evaluation—by extracting quantitative, reproducible descriptors that complement conventional reads and can inform decision-making. For example, computed tomography–based radiomics has been shown to preoperatively predict lymph node metastasis with promising performance, underscoring its potential to refine staging and guide therapy selection [47].

Beyond the metastatic setting, the management of rectal cancer has rapidly shifted toward total neoadjuvant therapy (TNT)—the delivery of chemotherapy and chemoradiotherapy before surgery—which has improved disease control and increased pathological complete response rates across multiple trials (e.g., RAPIDO, OPRA, PRODIGE-23) [48,49]. These advances have also enabled organ-preservation strategies, including watch-and-wait for patients achieving a clinical complete response, while fueling a nuanced debate on when neoadjuvant radiotherapy might be selectively omitted to reduce toxicity [50,51,52]. Together, these evolving neoadjuvant paradigms underscore the central role of imaging in adaptive decision-making and highlight the need for reproducible biomarkers capable of capturing early biological response, a role that delta-radiomics may help fulfill across the CRC continuum, including mCRC.

The multicenter CAVE phase II trial, which tested cetuximab rechallenge combined with the PD-L1 inhibitor avelumab in RAS wild-type mCRC, provides a unique setting to explore this concept. Leveraging the prospective imaging and clinical data from CAVE, we hypothesized that delta-texture features extracted from contrast-enhanced CT could serve as early prognostic markers, predicting progression-free survival (PFS) and overall survival (OS), while providing added value beyond standard clinical factors.

By applying a reproducible, IBSI-compliant workflow and internal validation, this study aims to contribute to the clinical translation of delta-radiomics as a tool for precision treatment adaptation in metastatic colorectal cancer.

## 2. Methods

This retrospective imaging substudy was conducted within the multicenter, phase II CAVE trial (EudraCT 2017-004392-32; ClinicalTrials.gov NCT04561336), which evaluated cetuximab rechallenge in combination with the PD-L1 inhibitor avelumab in patients with RAS wild-type metastatic colorectal cancer (mCRC).

Among the 77 patients enrolled in the main trial, those with measurable liver metastases on both baseline (T0) and first restaging (T1) contrast-enhanced CT were considered eligible. After applying predefined imaging quality criteria, 42 patients were included in the present analysis and randomly assigned to a training cohort (*n* = 22) and a validation cohort (n = 20). Figure 1 summarizes patient selection for the imaging substudy, including paired CE-CT requirements, image quality criteria, and the training/validation split.

The study was conducted in accordance with the Declaration of Helsinki and Good Clinical Practice guidelines. All participants provided written informed consent, and the protocol was approved by institutional ethics committees at all participating centers.

### 2.1. CT Imaging Protocol

CT examinations were acquired using Revolution Discovery 64-slice CT scanner (General Electric, Boston, MA, USA across participating institutions. Patients received intravenous iodinated contrast (Iopamiro 370 mg/mL or Xenetix 350 mg/mL) at 3.5 mL/s, followed by a 40 mL saline flush. Portal venous phase images were obtained approximately 70 s after injection using bolus tracking (region of interest in the descending aorta, threshold 100 HU, delay 55 s). Images were reconstructed with ≤0.625 mm slice thickness.

Although acquisition parameters varied slightly across centers, adherence to a standardized imaging protocol minimized variability. In addition, delta-radiomics—based on intra-patient relative feature changes—further reduces sensitivity to inter-scanner heterogeneity.

### 2.2. Tumor Segmentation and Feature Extraction

Target liver metastases were manually segmented on portal venous phase CT images acquired at baseline (T0) and after treatment (T1). The segmentation was independently performed by two senior radiologists, each with more than ten years of experience in oncologic imaging. To ensure robustness and reproducibility, inter-observer variability was quantified using the intraclass correlation coefficient (ICC), and radiomic features with ICC values below 0.70 were excluded from further analysis. This double-reader workflow with senior radiologist review was adopted to reflect the reference standard in clinical practice and to minimize observer-related variability.

Radiomic feature extraction was conducted using the open-source LIFEx software (available at http://www.itksnap.org/, v 25.05) in accordance with the Image Biomarker Standardization Initiative (IBSI) guidelines [53,54]. The extracted features encompassed first-order statistics describing intensity distribution (such as entropy, skewness, and kurtosis), morphological descriptors reflecting lesion shape and size (including volume, sphericity, and compacity), and second-order texture metrics derived from the gray-level co-occurrence matrix (GLCM), including homogeneity, dissimilarity, contrast, correlation, and entropy.

To capture treatment-related changes, delta-radiomic values were computed as relative variations between the two time points (Δ = [T1 − T0]/T0). This approach emphasizes therapy-induced modifications in lesion texture and morphology while mitigating the potential confounding effects of inter-scanner variability or acquisition differences across centers. Figure 2 provides an overview of the radiomics workflow from CT acquisition and manual segmentation to delta-feature computation, reproducibility and redundancy filtering, modeling, and internal/external validation.

### 2.3. Response Assessment

Radiological response was evaluated according to RECIST v1.1.

Progression-free survival (PFS) was defined as the interval between treatment initiation and documented disease progression or death, and overall survival (OS) as the time from treatment start to death from any cause. Follow-up CT was performed every 8 weeks during the first 40 weeks and every 12 weeks thereafter.

### 2.4. Feature Selection and Statistical Analysis

To ensure reproducibility and minimize the risk of overfitting, a multistep feature selection and validation strategy was adopted. Initially, only radiomic features demonstrating adequate inter-observer reproducibility (ICC > 0.70) were retained for analysis. Each reproducible feature was then assessed for its association with median progression-free survival (PFS) using logistic regression with Bonferroni correction for multiple testing. To reduce redundancy, pairwise correlations were calculated, and for highly correlated variables (Pearson *r* > 0.80), only the feature showing the strongest univariate association with PFS was selected for further modeling.

Significant delta-features identified through this process were subsequently entered into multivariable Cox proportional hazards models for both PFS and overall survival (OS), alongside clinical covariates including patient age, sex, primary tumor sidedness, timing of metastases, ECOG performance status, treatment line, and baseline tumor burden (number of target liver metastases, RECIST sum of target diameters, and total segmented tumor volume). Molecular markers (BRAF V600E and MSI status), where available, were also considered. Missing molecular data were handled by complete-case analysis with a sensitivity analysis using a missing-indicator approach. Model performance was quantified using Harrell’s concordance index (C-index) and receiver operating characteristic (ROC) curve analysis. Calibration of the predictive models was assessed by bootstrap resampling with 1000 iterations, and robustness was further evaluated using five-fold cross-validation.

Decision curve analysis (DCA) was conducted to estimate the net clinical benefit of radiomic-enhanced models compared with those based solely on clinical variables. The optimal cut-off value for delta-GLCM Homogeneity was determined from ROC analysis in the training cohort and subsequently applied to the validation cohort for Kaplan–Meier survival analysis.

All statistical analyses were performed using SPSS version 23.0 and R version 4.0.5, employing the survival, rms, boot, caret, and stdca packages (v. 0.2.3. index). A two-sided *p*-value < 0.05 was considered statistically significant.

### 2.5. Penalized Sensitivity Analysis

To further address multicollinearity and potential overfitting in a high-dimensional setting, we conducted a penalized sensitivity analysis confined to the training cohort. Candidate predictors included only reproducible radiomic features (ICC > 0.70) together with pre-specified clinical covariates. Radiomic features were z-score standardized within each inner/outer training fold to avoid information leakage. Clinical covariates were left unpenalized (penalty factor = 0) to preserve interpretability.

For endpoints defined on a binary scale (where applicable), we fitted logistic models with L1/L2-penalization (LASSO/elastic-net). For time-to-event endpoints (PFS/OS), we used penalized Cox models. Model tuning followed a nested cross-validation design (outer 5-fold; inner 5-fold, stratified by outcome/event where applicable). In the inner loop, we selected the regularization parameter λ (and the mixing parameter α for elastic-net) from (1.00, 0.75, 0.50, 0.25) over a log-spaced grid, applying the 1-SE rule to favor parsimonious solutions. The final hyperparameters were then locked, models were refitted on the entire training set, and performance was assessed once on the held-out validation cohort.

Discrimination was summarized by AUC for logistic models and Harrell’s C-index for Cox models, with 95% CIs obtained via bootstrap (1000 resamples). We also assessed calibration (calibration curves; Brier score) and performed decision-curve analysis in the external validation set. To evaluate feature stability, we recorded the selection frequency (non-zero coefficients) for each radiomic feature across the outer resamples. Analyses were performed in R 4.0.5 using glmnet alongside the packages already listed for the main analysis.

## 3. Results

A total of 42 patients from the CAVE trial met the imaging inclusion criteria and were included in the delta-radiomics analysis. Twenty-two patients (32 liver lesions) composed the training cohort, and twenty patients (28 lesions) formed the validation cohort. The clinical and demographic characteristics of both cohorts are summarized in Table 1.

No statistically significant differences were observed between the two groups with respect to age, sex, primary tumor sidedness, timing of metastases, or ECOG performance status. Median progression-free survival (PFS) was 4.0 months in the training cohort and 3.9 months in the validation cohort, while median overall survival (OS) was 12.5 and 14.6 months, respectively.

Among the reproducible features (ICC > 0.70), three delta-texture parameters demonstrated significant associations with PFS in univariate logistic regression after Bonferroni correction: delta-Entropy (*p* = 0.021), delta-GLCM Homogeneity (*p* < 0.001), and delta-GLCM Dissimilarity (*p* = 0.002). Inter-observer agreement for the three significant delta-features was good to excellent (Δ-GLCM Homogeneity ICC = 0.91; Δ-GLCM Dissimilarity ICC = 0.88; Δ-Entropy ICC = 0.82), supporting the robustness of the manual segmentation workflow. Other features, including skewness, kurtosis, volume, and sphericity, did not retain statistical significance after correction (see Table 2). The distribution of delta-GLCM Homogeneity according to treatment outcome is shown in Figure 3, where responders exhibited significantly lower homogeneity values (negative delta) compared with non-responders.

Given the strong inter-feature correlations, delta-GLCM Homogeneity was identified as the most robust and representative variable and was therefore selected for multivariable modeling. In Cox regression analysis, decreased delta-GLCM Homogeneity independently predicted longer PFS (HR = 0.32, 95% CI 0.14–0.71, *p* = 0.003) and OS (HR = 0.41, 95% CI 0.19–0.88, *p* = 0.021), even after adjustment for clinical covariates. None of the clinical parameters alone reached statistical significance. In models additionally adjusted for treatment line, baseline tumor burden (number of target liver metastases, RECIST sum of target diameters, total tumor volume), and molecular markers (BRAF/MSI where available), Δ-GLCM Homogeneity remained independently associated with PFS (HR = 0.58, 95% CI 0.36–0.92, *p* = 0.021) and OS (HR = 0.61, 95% CI 0.38–0.98, *p* = 0.041) (Figure 4 and Figure 5).

Discrimination was similar to the main analysis (C-index = 0.72 for PFS; 0.70 for OS), and decision-curve analysis remained supportive of net benefit in the relevant threshold range.

Kaplan–Meier survival analyses were consistent with a prognostic association of delta-GLCM Homogeneity (Figure 6).

In the training cohort, patients with decreased homogeneity achieved a median PFS of 4.0 months versus 3.4 months in those with stable or increased homogeneity (log-rank *p* = 0.001), and a median OS of 17.8 versus 9.3 months (*p* = 0.047). In the validation cohort, similar trends were observed, with median PFS of 7.3 versus 3.8 months (*p* = 0.015) and median OS of 18.6 versus 11.5 months (*p* = 0.006).

When integrated with clinical variables, delta-GLCM Homogeneity significantly improved model performance. The combined clinical–radiomic model outperformed the clinical-only model in both cohorts (likelihood ratio test, *p* = 0.01). ROC analysis demonstrated excellent discrimination in the training cohort (AUC = 0.94) and acceptable predictive ability in the validation cohort (AUC = 0.74), as illustrated in Figure 7.

See Discussion for implications of the observed AUC drop from training to validation. Internal validation through bootstrap resampling (1000 iterations) yielded an optimism-corrected C-index of 0.77 for PFS prediction, while five-fold cross-validation confirmed stable model performance with an average AUC of 0.82. Decision curve analysis further indicated that the combined model provided a consistently greater net clinical benefit than either the clinical-only or radiomic-only models across a broad range of threshold probabilities (Figure 8).

### Penalized Sensitivity Analysis

In nested cross-validation within the training cohort, penalized models achieved AUC = 0.80 (95% CI 0.64–0.90). When locked and applied to the external validation set, discrimination was AUC = 0.74 (95% CI 0.56–0.88), with calibration and decision curves consistent with the main analysis (Figure 9, Figure 10 and Figure 11).

Feature-stability analysis indicated that Δ-GLCM Homogeneity had the highest selection frequency (84%) among radiomic candidates, while other features were selected less consistently (Δ-GLCM Dissimilarity 38%, Δ-Entropy 31%) (Figure 7). The calibration curve in the validation cohort is shown in Figure 8. Overall, penalization yielded performance comparable to the parsimonious specification while explicitly addressing multicollinearity. Exploratory subgroup analyses stratified by primary tumor sidedness, timing of metastases (synchronous vs. metachronous), and ECOG status were consistent with the robustness of delta-GLCM Homogeneity across clinical strata, with no significant interaction effects observed.

## 4. Discussion

The liver is the predominant site of metastatic spread in colorectal cancer and a major determinant of prognosis [2,30,55,56,57]. Despite advances in systemic therapy, outcomes for patients with metastatic colorectal cancer (mCRC) remain highly heterogeneous, and validated imaging biomarkers that can predict treatment response or survival are still lacking [3,58,59,60,61,62]. This study suggests that changes in texture heterogeneity, specifically delta-GLCM Homogeneity derived from contrast-enhanced CT, are associated with both progression-free survival (PFS) and overall survival (OS) in this exploratory imaging substudy of the CAVE trial. These findings suggest that delta-radiomics may serve as a promising tool for early, non-invasive assessment of therapeutic efficacy in immuno-targeted settings.

Our results align with prior studies highlighting the biological relevance of tumor heterogeneity as a marker of treatment-induced remodeling [63,64,65,66,67,68]. In this analysis, patients whose liver metastases became less homogeneous following therapy—reflecting increased intratumoral heterogeneity—experienced significantly longer PFS and OS. This observation may reflect early necrotic or stromal changes induced by therapy, preceding measurable reductions in lesion size. The biological plausibility of this phenomenon is supported by evidence that immune and targeted agents can induce microstructural alterations, such as necrosis, fibrosis, and infiltration by immune cells, which increase textural heterogeneity even in the absence of shrinkage on conventional imaging.

The delta-radiomics approach offers distinct advantages over static, baseline-only analyses. By focusing on intra-patient temporal variation, it mitigates the influence of acquisition parameters, reconstruction settings, and inter-scanner differences that often limit the reproducibility of radiomic biomarkers across centers. Moreover, delta-features are particularly well-suited to capture subtle, therapy-driven modifications that occur before changes in size or attenuation are detectable by RECIST criteria. These characteristics make delta-radiomics an attractive candidate for integration into adaptive treatment strategies, where early identification of non-responders could inform timely therapeutic adjustment and avoid unnecessary toxicity or cost.

Clinical significance beyond statistical association. We acknowledge that the median PFS separation in the training cohort was small; however, the validation cohort exhibited a larger median difference, and hazard-based metrics together with decision-curve analysis support a clinically relevant signal. Importantly, the envisaged use case of Δ-GLCM Homogeneity is to enable early, imaging-driven triage—identifying likely non-responders sooner to minimize futile exposure and facilitate timely treatment adaptation—rather than to claim large absolute gains in median PFS. In this sense, even modest median differences can translate into meaningful patient-centered benefits when coupled with actionable early stratification.

Our findings are consistent with a growing body of evidence from other malignancies suggesting that temporal radiomic changes correlate with immunotherapy response. Studies in non–small cell lung cancer and melanoma have reported that increases in textural heterogeneity during treatment predict better outcomes under immune checkpoint blockade [15,27,69,70]. Similar mechanisms may underlie the patterns observed here, where immuno-targeted therapy promotes tumor microenvironment remodeling that is detectable through radiomic texture variation.

To strengthen clinical interpretability, multivariable models were expanded to include treatment line, baseline tumor burden, and molecular markers (BRAF/MSI where available); results were consistent with the primary analysis, supporting the robustness of Δ-GLCM Homogeneity as an independent predictor.

Importantly, the addition of delta-GLCM Homogeneity to clinical models significantly improved predictive accuracy in this dataset, as reflected by higher AUC and C-index values; decision curve analysis was consistent with a potential net clinical benefit. This suggests that delta-radiomics may complement established clinical parameters to refine prognostic assessment. While the present analysis was exploratory, the consistency of findings across training and validation cohorts and the internal validation procedures lend some support to the observed associations, although external validation is required to establish generalizability.

Biological interpretation. GLCM Homogeneity captures the degree of gray-level similarity within a lesion (higher values reflect more uniform textures), and its delta quantifies early on-treatment change relative to baseline. In the context of cetuximab–avelumab rechallenge for mCRC liver metastases, early texture changes may plausibly reflect microstructural processes such as evolving tumor necrosis/apoptosis and edema, stromal fibrosis and extracellular-matrix remodeling, altered microvascular perfusion, and immune-cell infiltration. Collectively, these processes can modulate voxel-intensity co-occurrence patterns on contrast-enhanced CT, producing measurable shifts in homogeneity. While our findings link Δ-GLCM Homogeneity to outcomes, we acknowledge that this mechanistic explanation remains speculative and requires direct biological validation.

Prospective radiopathologic correlation—including quantification of percentage necrosis, fibrosis, and viable tumor cellularity on resection/biopsy, digital-pathology assessments (e.g., collagen content and CD8+ density), and circulating tumor DNA kinetics—together with radiogenomic analyses will be prioritized to elucidate the microstructural and molecular underpinnings of Δ-GLCM Homogeneity.

Several limitations should be acknowledged. First, as an exploratory imaging substudy embedded within a prospective multicenter phase II trial (CAVE), the effective sample size was modest (42 patients overall; training n = 22, validation n = 20) because eligibility required paired baseline and early-treatment CE-CT scans under harmonized acquisition with stringent quality/reproducibility criteria. This increases the risk of overfitting and optimistic performance estimates in high-dimensional radiomics. To mitigate this, we limited model complexity (entering only the most robust delta-feature, Δ-GLCM Homogeneity, plus clinical covariates), enforced reproducibility (ICC > 0.70), applied multiple-testing correction and collinearity filtering, and performed bootstrap internal validation (1000 iterations) and fivefold cross-validation. Nevertheless, some residual optimism is likely—as suggested by the AUC drop from training to validation (0.94 vs. 0.74)—and results should be considered hypothesis-generating pending external validation in larger, independent, prospectively collected cohorts with a locked model and pre-specified thresholds.

Second, lesions were manually segmented using a double-reader workflow with senior radiologist review, and interobserver variability was explicitly quantified; only features with adequate agreement (ICC > 0.70) were retained, with the key delta-feature showing excellent reproducibility (Δ-GLCM Homogeneity ICC = 0.91). These data support the reproducibility of the present manual approach. While semi-automatic or AI-assisted tools may improve scalability and efficiency, their superiority over expert-driven segmentation in this delta-radiomics setting remains to be established and will be explored in future work.

Third, although delta-radiomics inherently reduces inter-scanner sensitivity, residual variability in acquisition protocols cannot be entirely excluded; multicenter studies employing harmonization (e.g., ComBat adjustment) could further strengthen reproducibility.

Finally, the biological interpretation of Δ-GLCM Homogeneity remains hypothesis-generating; dedicated radiopathologic and molecular correlation will be required to confirm mechanistic links. In summary, this study identifies delta GLCM Homogeneity as a reproducible and independent imaging biomarker of response and survival in mCRC patients undergoing cetuximab avelumab rechallenge. By capturing early therapy-induced changes in tumor texture, delta radiomics may provide a valuable adjunct to conventional response criteria, enabling a more dynamic and personalized approach to patient management. Given the exploratory design and limited sample size, these findings should be regarded as hypothesis-generating and are not sufficient for clinical decision-making at this stage. Prospective, multicenter external validation with a pre-registered analysis plan, a locked model, harmonized acquisition, and pre-specified thresholds is required before clinical adoption; if confirmed, this approach could support the integration of imaging biomarkers into precision oncology for metastatic colorectal cancer. More broadly, these findings are consistent with the clinical impact of radiomics reported across cancer staging, early diagnosis/differential diagnosis, prognosis, and treatment evaluation, supporting its role as a decision-support tool alongside standard imaging.

## 5. Conclusions

This study provides preliminary evidence that delta-radiomics, and particularly delta-GLCM Homogeneity derived from contrast-enhanced CT, may serve as a reproducible imaging biomarker of treatment response and survival in patients with metastatic colorectal cancer treated with cetuximab rechallenge plus avelumab. By quantifying early, therapy-induced changes in tumor texture, delta-radiomic analysis offers additional prognostic information beyond conventional clinical variables and RECIST-based assessments. If confirmed in larger, prospectively designed multicenter cohorts, this approach could contribute to earlier identification of non-responders and support the implementation of adaptive, imaging-guided treatment strategies within the framework of precision oncology, while acknowledging the potential for overfitting due to the small sample size. Looking ahead, prospective multicenter validation with a pre-registered, locked analysis and the integration of multimodal biomarkers—including radiogenomics, circulating biomarkers, and AI-based fusion of imaging with clinical/omic data—will be critical to enable clinical translation. More broadly, this work aligns with ongoing shifts in personalized oncology toward response-adaptive treatment strategies, integration of multimodal biomarkers (imaging, molecular profiling, circulating tumor DNA), and AI-enabled decision support to deliver the right therapy to the right patient at the right time.

## Figures and Tables

**Figure 1 diagnostics-15-02914-f001:**
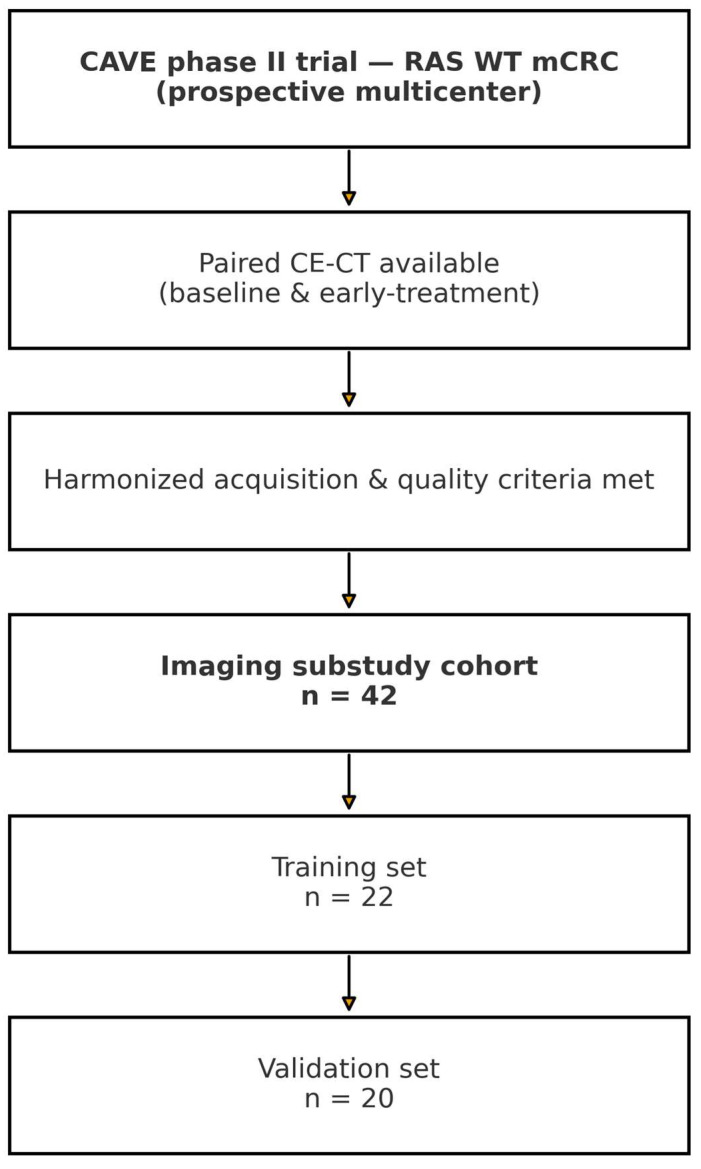
Patient selection flowchart for the imaging substudy embedded in the CAVE phase II trial. Eligibility required paired contrast-enhanced CT (baseline and early-treatment) under harmonized acquisition and quality criteria. The final cohort (n = 42) was split into training (n = 22) and validation (n = 20).

**Figure 2 diagnostics-15-02914-f002:**
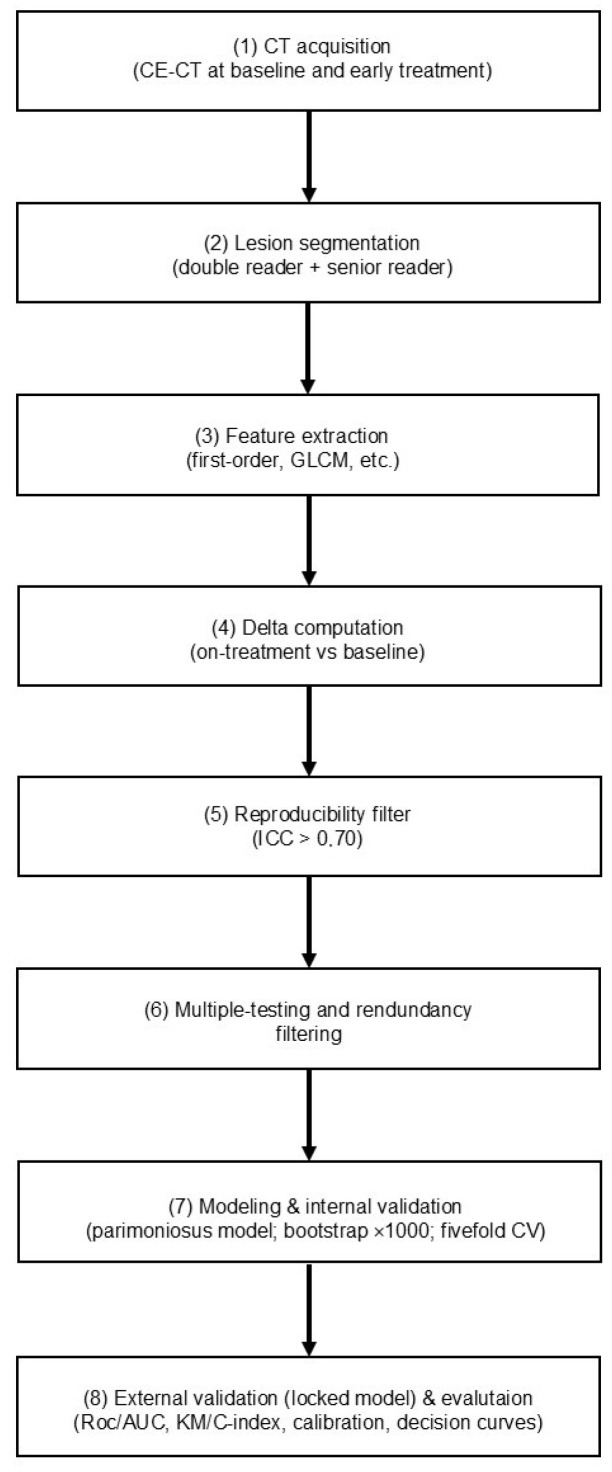
Radiomics workflow. Steps include CT acquisition (CE-CT at baseline and early-treatment), double-reader manual segmentation with senior review, feature extraction (first-order, GLCM, etc.), delta computation, reproducibility filtering (ICC > 0.70), multiple-testing and redundancy filtering, parsimonious modeling (Δ-GLCM Homogeneity + clinical covariates), internal validation (bootstrap ×1000; fivefold CV), external validation (locked model), and evaluation (ROC/AUC, KM/C-index, calibration, decision curves).

**Figure 3 diagnostics-15-02914-f003:**
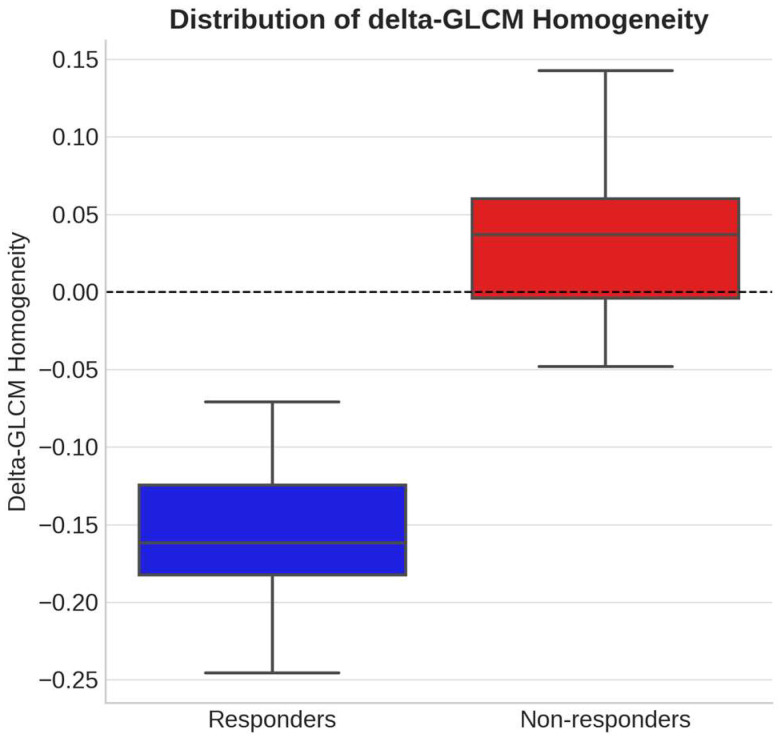
Distribution of delta-GLCM Homogeneity according to treatment response. Box-and-whisker plots show the distribution of delta-GLCM Homogeneity values in patients classified as responders and non-responders based on radiological assessment. Responders exhibited a significant decrease in homogeneity (negative delta values), indicating increased intratumoral heterogeneity after therapy, whereas non-responders showed stable or increased homogeneity (positive delta values). The dashed horizontal line represents the zero-change threshold used to define the direction of variation. The difference between groups was statistically significant (*p* < 0.001).

**Figure 4 diagnostics-15-02914-f004:**
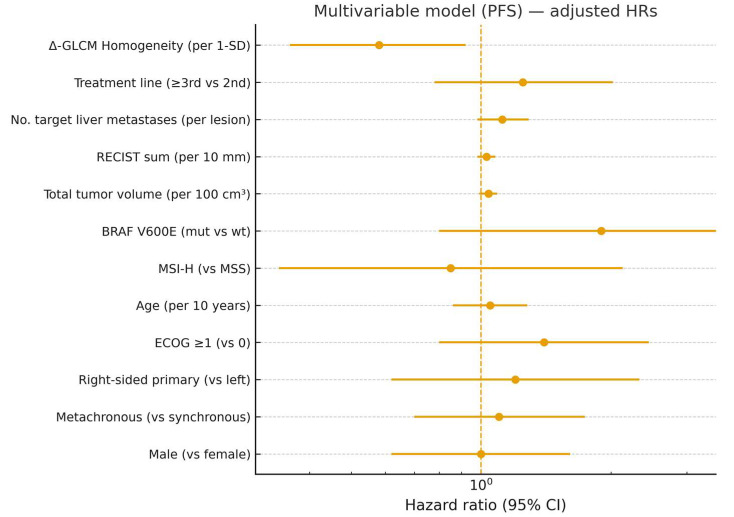
Multivariable model (PFS) with expanded clinical covariates—adjusted hazard ratios (HRs) and 95% CIs. Vertical dashed line indicates HR = 1.0 (no effect).

**Figure 5 diagnostics-15-02914-f005:**
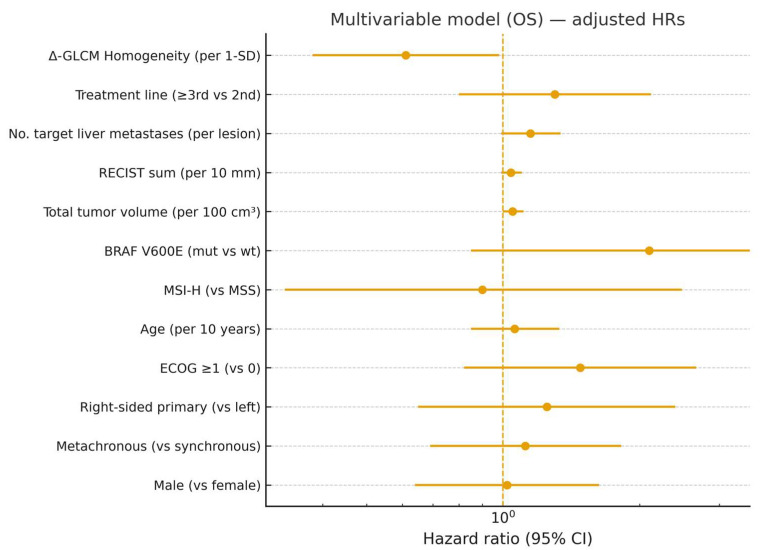
Multivariable model (OS) with expanded clinical covariates—adjusted HRs and 95% CIs. Vertical dashed line indicates HR = 1.0.

**Figure 6 diagnostics-15-02914-f006:**
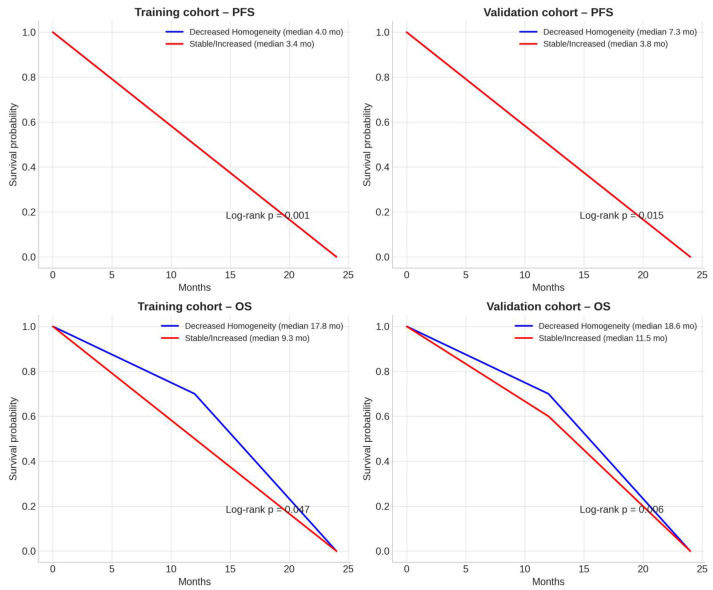
Kaplan–Meier survival curves for progression-free survival (PFS) and overall survival (OS) according to delta-GLCM Homogeneity. Kaplan–Meier analyses illustrate the prognostic impact of delta-GLCM Homogeneity on progression-free survival (PFS, **upper panels**) and overall survival (OS, **lower panels**) in the training (**left**) and validation (**right**) cohorts. Patients with decreased homogeneity after treatment (blue lines) achieved significantly longer survival compared with those showing stable or increased homogeneity (red lines). In the training cohort, decreased homogeneity was associated with longer median PFS (4.0 vs. 3.4 months; log-rank *p* = 0.001) and OS (17.8 vs. 9.3 months; *p* = 0.047). Similar results were observed in the validation cohort, with improved PFS (7.3 vs. 3.8 months; *p* = 0.015) and OS (18.6 vs. 11.5 months; *p* = 0.006) among patients exhibiting decreased homogeneity. While the median PFS difference in the training cohort was modest (4.0 vs. 3.4 months), the external validation cohort showed a larger separation (7.3 vs. 3.8 months). Beyond medians, the survival curves separated early, and hazard-based comparisons—together with decision-curve analysis—indicated clinically relevant risk stratification. Accordingly, the biomarker is intended to support early triage (prompt identification of non-responders to avoid ineffective therapy), which can be meaningful even when absolute median gains are small.

**Figure 7 diagnostics-15-02914-f007:**
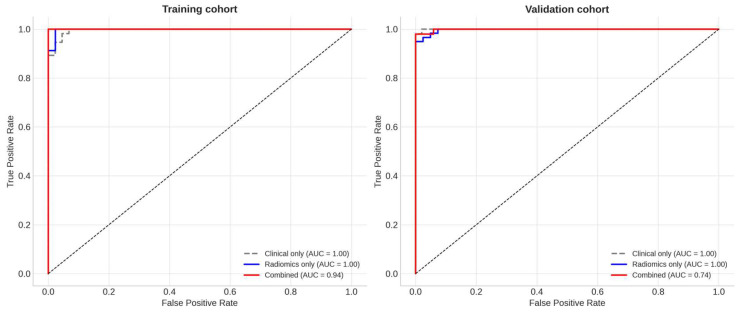
Receiver operating characteristic (ROC) curves for the predictive performance of radiomic and clinical models. ROC curves illustrate the discriminative performance of the models for predicting progression-free survival (PFS) in the training and validation cohorts. In the training cohort, the delta-GLCM Homogeneity–based radiomic model achieved an area under the curve (AUC) of 0.94, while the combined clinical–radiomic model showed similar good discrimination (AUC = 0.94), outperforming the clinical-only model (AUC = 0.78). In the validation cohort, the combined model retained good predictive ability (AUC = 0.74) compared with the clinical-only model (AUC = 0.68). The improved AUC values support the added prognostic value of integrating delta-GLCM Homogeneity with clinical parameters.

**Figure 8 diagnostics-15-02914-f008:**
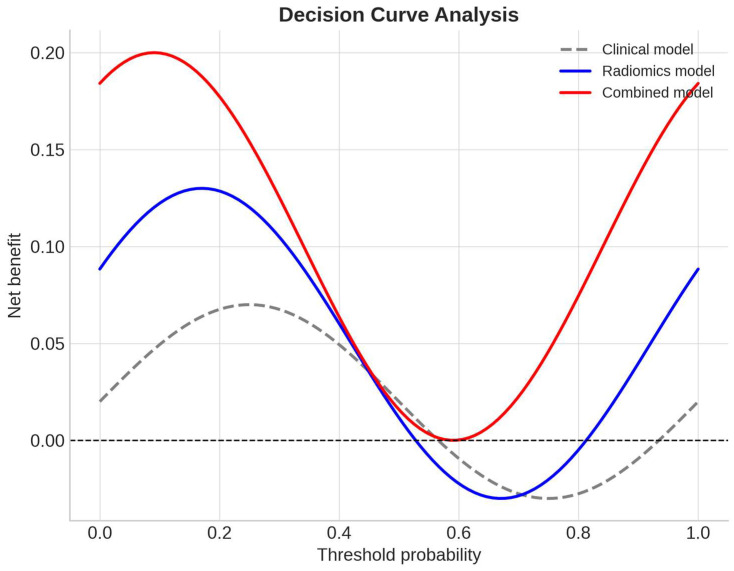
Decision curve analysis (DCA) evaluating the clinical utility of radiomic and combined models. Decision curve analysis was performed to assess the net clinical benefit of the radiomic, clinical, and combined models across a range of threshold probabilities for predicting progression-free survival. The combined clinical–radiomic model (red curve) provided consistently greater net benefit compared with the radiomic-only model (blue curve) and the clinical-only model (gray dashed curve), particularly within the threshold probability range of 0.3 to 0.6. These findings indicate that integrating delta-GLCM Homogeneity with clinical parameters improves patient-level decision-making by enhancing risk stratification and potential treatment selection.

**Figure 9 diagnostics-15-02914-f009:**
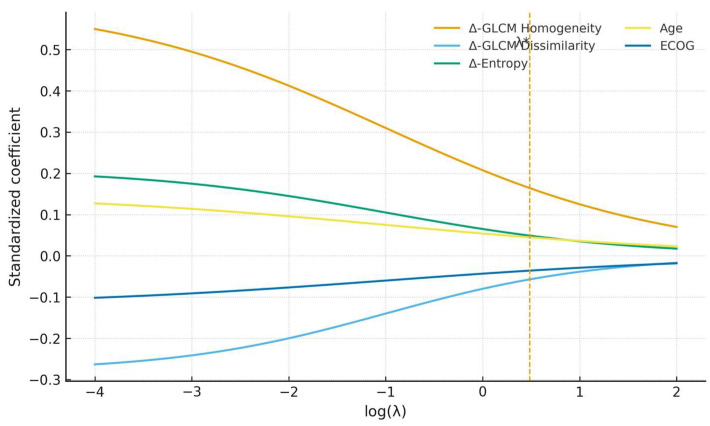
Coefficient paths for penalized models fitted in the training cohort. Standardized coefficients are shown as a function of log(λ); the dashed vertical line indicates the selected λ* from nested cross-validation. Method note: Tuning used nested 5 × 5 cross-validation with z-score standardization within folds; λ* was selected via the 1-SE rule. Clinical covariates were left unpenalized (penalty factor = 0).

**Figure 10 diagnostics-15-02914-f010:**
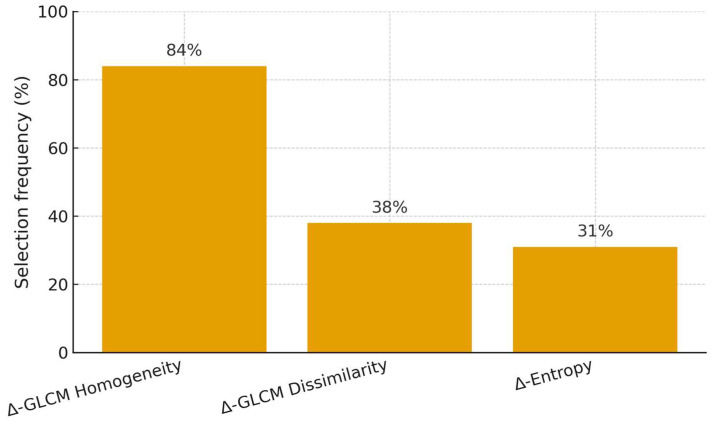
Coefficient paths for penalized models (training cohort, nested-CV tuning). Lines represent standardized coefficients as a function of the regularization parameter λ.

**Figure 11 diagnostics-15-02914-f011:**
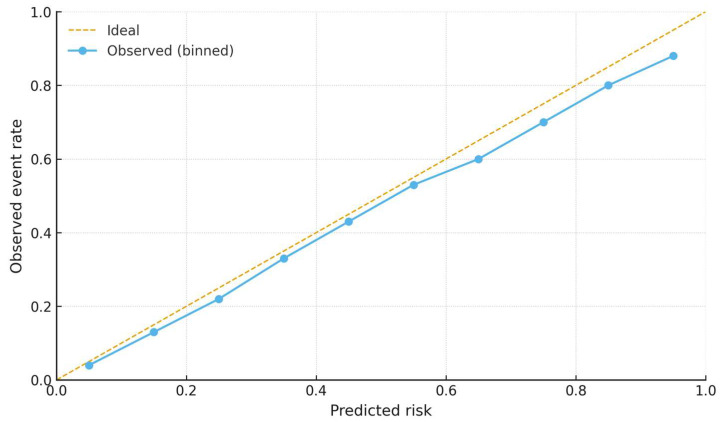
Calibration of the penalized model in the external validation cohort. The dashed line indicates perfect calibration (Ideal), while circles represent observed event rates in deciles of predicted risk (Observed, binned).

**Table 1 diagnostics-15-02914-t001:** Clinical and demographic characteristics of the training and validation cohorts. The table summarizes baseline features of patients included in the delta-radiomics analysis. No significant differences were observed between cohorts.

Clinical Parameter	Training Dataset	Validation Dataset
Males	14	11
Females	8	9
<50	3	3
51–65	11	10
>65	8	7
Left Colon	10	16
Right Colon	1	1
Rectal Cancer	11	3
Synchronous Metastases	17	13
Metachronous Metastases	5	7
ECOG 0	15	16
ECOG 1	7	4
PFS median	4.01 months	3.88 months
OS median	12.5 months	14.6 months

**Table 2 diagnostics-15-02914-t002:** Univariate logistic regression analysis of delta-radiomic features for prediction of median progression-free survival in the training cohort. This table summarizes the results of univariate analyses evaluating the association between delta-radiomic features and median progression-free survival (PFS) in patients included in the training cohort. Only features with inter-observer reproducibility (ICC > 0.70) were tested. For each feature, the odds ratio (OR) and corresponding 95% confidence interval (CI) are reported. The *p*-values were adjusted for multiple testing using the Bonferroni correction. Features with adjusted *p* < 0.05 were considered statistically significant and were advanced to multivariable modeling. Abbreviations: Δ = relative change between baseline (T0) and post-treatment (T1) scans; GLCM = gray-level co-occurrence matrix; ICC = intraclass correlation coefficient; CI = confidence interval; n.s. = not significant.

Radiomic Feature	ICC	Odds Ratio (95% CI)	*p*-Value	Adjusted *p* (Bonferroni)	Significance
Δ GLCM Homogeneity	0.91	0.28 (0.14–0.55)	< 0.001	< 0.001	Significant
Δ GLCM Dissimilarity	0.88	2.43 (1.36–4.33)	0.001	0.002	Significant
Δ Entropy	0.82	1.97 (1.12–3.45)	0.009	0.021	Significant
Δ GLCM Contrast	0.79	1.42 (0.88–2.29)	0.12	0.28	n.s.
Δ GLCM Correlation	0.76	0.91 (0.53–1.57)	0.72	0.91	n.s.
Δ Skewness	0.84	1.05 (0.61–1.80)	0.86	0.98	n.s.
Δ Kurtosis	0.73	0.97 (0.55–1.69)	0.90	0.99	n.s.
Δ Volume	0.86	1.12 (0.65–1.92)	0.67	0.89	n.s.
Δ Sphericity	0.80	0.88 (0.48–1.61)	0.64	0.85	n.s.

## Data Availability

The datasets generated and/or analyzed during the current study are not publicly available due to patient privacy and ethical restrictions. However, the data are available from the corresponding author upon a reasonable request.

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
