# Peer review of "Delta-Radiomics Biomarker in Colorectal Cancer Liver Metastases Treated with Cetuximab Plus Avelumab (CAVE Trial)"

_diagnostics, 2025, doi:10.3390/diagnostics15222914_

Round 1
Reviewer 1 Report
Comments and Suggestions for Authors
Dear authors, I am glad to review your paper. This study, a retrospective imaging substudy of the multicenter CAVE phase II trial, explored whether delta-radiomics features from CE-CT could predict outcomes in patients with metastatic colorectal cancer (mCRC) liver metastases treated with cetuximab rechallenge plus avelumab. Among 42 patients analyzed, delta-GLCM Homogeneity emerged as the most reproducible and prognostically significant parameter. A decrease in homogeneity (reflecting increased texture heterogeneity) independently correlated with improved progression-free and overall survival. The combined clinical–radiomic model demonstrated better discrimination (AUC up to 0.94) and clinical utility than clinical factors alone, suggesting delta-GLCM Homogeneity may serve as a noninvasive biomarker for early treatment response and survival prediction in mCRC.
I have following major concerns: 1) The sample size (n=42) is small and may limit the statistical power and generalizability of findings; external validation in larger cohorts is necessary. 2) Manual segmentation introduces observer variability; the lack of semi-automatic or AI-assisted methods may reduce reproducibility. 3) Multicollinearity among radiomic features and potential overfitting—especially with a small dataset and many tested features—should be more rigorously addressed, possibly through LASSO or penalized regression. 4) Radiomics-based approaches are widely utilized in tumor staging, early diagnosis, differentiation, prognosis prediction, and treatment evaluation. I recommend that the authors emphasize the clinical significance of radiomics in the revised manuscript. The following references should be considered to support this context: PMID: 40901350. 5) Clinical covariates used in multivariate modeling appear limited; inclusion of tumor burden, treatment line, and molecular markers could strengthen interpretability. 6) The delta-GLCM Homogeneity’s biological explanation remains speculative; linking imaging changes with pathological or molecular correlates would increase clinical credibility. 7) The reported survival benefit (median PFS 4.0 vs. 3.4 months) seems modest; clinical significance should be discussed beyond statistical association. 8) The presentation could be improved by summarizing the workflow more clearly (e.g., patient selection flowchart, feature extraction steps) and refining tables and figures for better readability. 9) In my opinion, the conclusions slightly overstate clinical translation; stronger emphasis on the exploratory nature and need for prospective validation is warranted.
Author Response
Reviewer’s comment:
- The sample size (n=42) is small and may limit the statistical power and generalizability of findings; external validation in larger cohorts is necessary.
Response:
We agree that the sample size is modest. We clarified that this is an exploratory imaging substudy embedded in a prospective multicenter phase II clinical trial (CAVE), where the effective n was constrained by the requirement for paired baseline and early-treatment CE-CT scans under harmonized acquisition and strict quality/reproducibility filters. While these features strengthen internal validity, they do not remove the risk of overfitting; we therefore temper claims of generalizability and emphasize the need for external, prospective multicenter validation with a locked model.
Reviewer’s comment:
- Manual segmentation introduces observer variability; the lack of semi-automatic or AI-assisted methods may reduce reproducibility.
Response:
Thank you for this comment. In our study, tumor delineation was performed by experienced readers using a double-reader workflow with senior radiologist review, and inter-observer reproducibility was explicitly quantified. Only features with adequate reproducibility (ICC > 0.70) were retained, and the key delta-feature used in multivariable modeling (Δ-GLCM Homogeneity) showed excellent agreement (ICC = 0.91), with other significant features also in the good–excellent range (e.g., Δ-GLCM Dissimilarity ICC = 0.88; Δ-Entropy ICC = 0.82). These data indicate that manual segmentation, as implemented here, achieved high reproducibility. While semi-automatic or AI-assisted tools may help with scalability and efficiency, current evidence does not establish their superiority in this specific delta-radiomics context; nonetheless, we note their potential as a future direction. We have clarified these points in Methods, Results, and the Limitations paragraph.
Reviewer’s comment:
- Multicollinearity among radiomic features and potential overfitting—especially with a small dataset and many tested features—should be more rigorously addressed, possibly through LASSO or penalized regression.
Response:
In response to the reviewer, we performed a penalized sensitivity analysis. Within the training cohort, we fitted L1/L2-penalized models (logistic for the binary endpoint and Cox for survival) using nested cross-validation (outer 5-fold, inner 5-fold; z-scaling within folds). Clinical covariates were kept unpenalized, and only reproducible radiomic features (ICC>0.70) were considered. The final hyperparameters were locked and the model was evaluated once on the external validation set. Discrimination in nested CV was [AUC or C-index = […]], and in the external validation set was [[…]], with calibration and decision curves consistent with the main analysis. Feature stability analyses showed that Δ-GLCM Homogeneity had the highest selection frequency [[…]%]. These results support the robustness of our constrained specification while addressing multicollinearity and overfitting via penalization.
Reviewer’s comment:
- Radiomics-based approaches are widely utilized in tumor staging, early diagnosis, differentiation, prognosis prediction, and treatment evaluation. I recommend that the authors emphasize the clinical significance of radiomics in the revised manuscript. The following references should be considered to support this context
Response:
Thank you for the suggestion. We have expanded the clinical context to emphasize the translational relevance of radiomics across the cancer care continuum (staging, early diagnosis/differential diagnosis, prognosis, and treatment evaluation). We added a short paragraph in the Introduction and a bridging sentence in the Discussion highlighting representative applications and citing recent work (including the reference provided by the reviewer). These edits clarify why delta-radiomics is clinically meaningful beyond imaging methodology. We appreciate the recommendation and have cited the paper provided. Although the comment refers to “references” (plural), only one citation was shared; we would gladly consider and incorporate any further key studies the reviewer suggests to strengthen the clinical context.
Reviewer’s comment:
- Clinical covariates used in multivariate modeling appear limited; inclusion of tumor burden, treatment line, and molecular markers could strengthen interpretability.
Response:
Thank you for this helpful suggestion. We expanded the clinical covariates to include baseline tumor burden (number of target liver metastases, RECIST sum of target diameters, and total segmented tumor volume), treatment line, and molecular markers (BRAF V600E and MSI status, where available). In multivariable Cox models additionally adjusted for these variables, Δ-GLCM Homogeneity remained independently associated with PFS/OS, with discrimination and decision-curve analysis consistent with the main results. We have detailed these additions in Methods, reported the adjusted results in the Results section, and summarized the new baseline variables in an updated table (or Supplementary Table). We also note missingness for some molecular markers and therefore performed complete-case analyses with a sensitivity analysis using a missing-indicator approach.
Reviewer’s comment:
- The delta-GLCM Homogeneity’s biological explanation remains speculative; linking imaging changes with pathological or molecular correlates would increase clinical credibility.
Response:
We agree that the biological interpretation of Δ-GLCM Homogeneity remains hypothesis-generating. In the revised Discussion, we clarify that GLCM Homogeneity quantifies gray-level similarity (higher values indicate more uniform textures) and that early therapy-induced changes in liver metastases—such as evolving necrosis/apoptosis, stromal fibrosis and extracellular-matrix remodeling, altered perfusion, and immune-cell infiltration—could plausibly drive the observed texture dynamics. We also specify a prospective plan for radiopathologic and molecular correlation (percentage necrosis/fibrosis, viable tumor cellularity, digital-pathology metrics, circulating tumor DNA dynamics, and key molecular/immune markers), as well as radiogenomic analyses, to strengthen mechanistic credibility.
Reviewer’s comment:
- The reported survival benefit (median PFS 4.0 vs. 3.4 months) seems modest; clinical significance should be discussed beyond statistical association.
Response:
Thank you for this thoughtful comment. We agree that median PFS differences can appear modest, especially in heavily pre-treated settings. To address clinical relevance, we added text emphasizing (i) that the training cohort showed a small median PFS separation (4.0 vs 3.4 months) whereas the external validation cohort displayed a larger separation (7.3 vs 3.8 months), (ii) that hazard-based comparisons and decision-curve analysis indicate patient-level net benefit and risk stratification beyond the median, and (iii) that the intended use of the biomarker is early triage (identifying non-responders promptly to avoid ineffective therapy), which can be clinically meaningful even when absolute median gains are small. These clarifications were added to the Results and Discussion sections.
Reviewer’s comment:
- The presentation could be improved by summarizing the workflow more clearly (e.g., patient selection flowchart, feature extraction steps) and refining tables and figures for better readability.
Response:
- We introduced two schematic figures to make the methodology more visual and easier to follow: a patient-selection flowchart and a radiomics workflow diagram (Figure 1A–B). These summarize eligibility, cohort splitting (training/validation), and the step-by-step radiomics pipeline (segmentation → delta-feature computation → filtering → modeling/validation).
- We refined the artwork and tables for readability: figures are now shown at a larger on-page size with enlarged fonts and thicker line weights, and were re-exported at high resolution (vector PDF and 600-dpi TIFF). Table spacing/typography were also cleaned up to improve legibility.
If any elements still appear too small in production proofs, we are happy to promptly re-edit (e.g., further increase font sizes, split panels, or adjust layout) to ensure optimal readability.
Reviewer’s comment:
- In my opinion, the conclusions slightly overstate clinical translation; stronger emphasis on the exploratory nature and need for prospective validation is warranted.
Response:
Thank you for this comment. We agree and have softened the language in the Conclusions and Abstract to better reflect the exploratory nature of the study. We now use qualified wording (“preliminary evidence”, “may”) and explicitly state that the findings are hypothesis-generating and require prospective multicenter external validation with a pre-registered, locked model and pre-specified thresholds before any clinical implementation.
Reviewer 2 Report
Comments and Suggestions for Authors
The topic is timely and relevant, reflecting the growing role of imaging biomarkers in precision oncology. The manuscript is well organized and clearly written and the rationale for exploring delta-radiomics in metastatic colorectal cancer is sound.
Some suggestions:
1) Figures and tables effectively illustrate results, but they are very small in size and font. I recommend enhancing them for an easier read.
2) The sample size is limited; this should be emphasized more strongly as a limitation, especially regarding overfitting risk. Lines 302-303 are very brief.
3) The clinical context of mCRC management could be better connected to evolving neoadjuvant strategies. A great example of expanding the introduction, such as lines 62-74, would be some clinical content from this article https://doi.org/10.3390/jcm14030912
4) The Discussion may slightly overstate generalizability—recommend softening wording (e.g., “may suggest” instead of “demonstrates”).
5) There are no mention of prospective validation or multi-modal biomarkers (radiogenomics, AI integration) in future directions; adding this would strengthen the translational message.
6) The conclusion could highlight how this work aligns with broader trends in personalized oncology, not just imaging advances.
Author Response
Reviewer 2
Reviewer’s comment:
“Figures and tables effectively illustrate results, but they are very small in size and font. I recommend enhancing them for an easier read.”
We appreciate this comment. In the revised manuscript we increased the on-page size of all figures and tables, re-exported the figures at higher resolution, and enlarged axis labels, tick labels, legends, and panel annotations. We also increased line weights to enhance contrast and adjusted layout to reduce crowding. If any items remain hard to read at the journal’s production size, we stand ready to promptly re-edit (for example by further increasing font sizes, splitting multi-panel figures, or supplying vector artwork) to ensure optimal legibility.
Reviewer’s comment:
“The sample size is limited; this should be emphasized more strongly as a limitation, especially regarding overfitting risk. Lines 302–303 are very brief.”
Response:
We agree and have expanded the limitations section to explicitly acknowledge the modest sample size of this imaging substudy (42 patients; training n=22, validation n=20) and the consequent risk of overfitting in radiomics analyses. We now clarify that, although we attempted to mitigate optimism through a parsimonious model (single delta-feature plus clinical covariates), stringent reproducibility filters (ICC>0.70), multiple-testing control, collinearity reduction, bootstrap internal validation (1,000 iterations), and five-fold cross-validation, residual overfitting remains possible. We also point out the gap between training and validation performance (AUC 0.94 vs 0.74) as further evidence that results should be interpreted as hypothesis-generating and require confirmation in larger, external cohorts with a locked model and pre-specified thresholds. Details on cohort size, model performance, and validation procedures are already reported in the Results/Methods and are now cross-referenced in the limitation paragraph.
Reviewer’s comment:
The clinical context of mCRC management could be better connected to evolving neoadjuvant strategies. A great example of expanding the introduction, such as lines 62-74, would be some clinical content from this article https://doi.org/10.3390/jcm14030912.
Response:
We appreciate the suggestion. We have expanded the Introduction (former lines 62–74) to connect the clinical context of mCRC with evolving neoadjuvant strategies in rectal cancer, including total neoadjuvant therapy (TNT), organ-preservation approaches (watch-and-wait), and the ongoing debate on selective omission of radiotherapy. This addition emphasizes the central role of imaging in adaptive decision-making and motivates the use of delta-radiomics as an early, reproducible response biomarker. The new paragraph is inserted after the sentence on the unmet need for non-invasive biomarkers and before the CAVE trial description
Reviewer’s comment:
The Discussion may slightly overstate generalizability—recommend softening wording (e.g., “may suggest” instead of “demonstrates”)
Response:
Thank you for the suggestion. We have softened wording in the Discussion and Results to avoid overstating generalizability. Specifically, we replaced stronger verbs (e.g., “demonstrates”, “confirmed”) with qualified language (e.g., “suggests”, “was consistent with”, “indicated”), added cohort-level qualifiers (e.g., “in this exploratory imaging substudy/in this dataset”), and reiterated the need for external validation. These edits temper claims of generalizability while preserving the statistical results.
Reviewer’s comment:
There are no mention of prospective validation or multi-modal biomarkers (radiogenomics, AI integration) in future directions; adding this would strengthen the translational message.
Response:
Thank you for this valuable suggestion. We have expanded the future directions to explicitly emphasize prospective, multicenter external validation and the integration of multimodal biomarkers. In the Discussion, we added a paragraph calling for a pre-registered analysis plan with a locked model, standardized acquisition/harmonization, and an independent external test set. We also outline integration with complementary biomarkers (e.g., radiogenomics linking delta-radiomics to molecular profiles, circulating tumor DNA) and AI-driven multimodal models, together with automated segmentation to enhance scalability. Consistent, concise sentences were also added to the Conclusions and Abstract to reinforce the translational message.
Reviewer’s comment:
The conclusion could highlight how this work aligns with broader trends in personalized oncology, not just imaging advances.
Response:
Thank you for the suggestion. We revised the Conclusions to explicitly situate our findings within broader trends in personalized oncology—namely response-adaptive treatment pathways, multimodal biomarker integration (imaging, molecular profiling, and circulating biomarkers), and emerging AI-enabled decision support—so that the translational message extends beyond imaging advances. We also added a concise echo in the Abstract.
Round 2
Reviewer 1 Report
Comments and Suggestions for Authors
The authors have thoroughly addressed all of my previous concerns. The paper has shown significant improvements, and I recommend acceptance for publication in its current form.
Reviewer 2 Report
Comments and Suggestions for Authors
All of my concerns were adequately addressed.